# Edu-Escape Rooms

Mario Grande-de-Prado [1,*], Sheila García-Martín [1], Roberto Baelo [1] and Víctor Abella-García [2]

1   Department of General and Specific Didactics and Educational Theory, University of León, 24071 León, Spain; sgarcm@unileon.es (S.G.-M.); rbaea@unileon.es (R.B.)
2   Department of Educational Sciences, University of Burgos, 09001 Burgos, Spain; vabella@ubu.es
*   Correspondence: mgrap@unileon.es

**Definition:** Escape Rooms are cooperative games in which players must find clues, solve puzzles, and perform a variety of tasks within a limited time. The goal is usually to escape or leave a room, place, or environment. When the Escape Rooms have a pedagogical purpose, they are usually called Edu-Escape Rooms and can be related to gamification and Game-Based Learning. The potential for student engagement and motivation is one of the main advantages of Edu-Escape Rooms.

**Keywords:** active methodologies; gamification; Game-Based Learning; educational games; Escape Room; breakout

## 1. Introduction

The European Higher Education Area advocates the implementation of active learning education methodologies complementary to traditional ones in order to face the new socio-educational context [1]. Active methodologies are one of the most interesting approaches to developing cooperative learning and student involvement in the classroom [2]. Everything related to games occupies a prominent place within these methodologies. Game is meaningful, spontaneous, and motivating [3,4]. In this regard, Piaget and Vygotsky [5,6] highlight the role of game in cognitive development, as it allows the incorporation of strategies, norms, and values in personal development.

Among the advantages offered by the games, it is worth highlighting their important didactic potential, which ranges from adapting to different learning rhythms, allowing mistakes, receiving instant feedback, and developing creativity to increasing the motivation and socialization of the students. Likewise, games enhance the student's commitment and participation in tasks as well as in the acquisition of skills [7–9]. Among the drawbacks of the educative use of the games is the risk of potential excessive competitiveness and inadequate time management; this, along with other particular aspects of each game, should be taken into account [10].

From an educational perspective, there are three important concepts linked to games: gamification, Game-Based Learning, and serious games. Although they are all related, they have different characteristics.

Gamification is the most known. It is often used to designate any activity in which playing and education or training are related; however, this concept is not precise. Gamification consists in the use of elements and mechanics of playing in non-playful contexts [11,12]. It is often implemented with help of online platforms, such as Classcraft or Classdojo [13,14]. Nowadays, gamification has become frequent inside educational research [15] and business training [16,17]. Very often this implementation consists in using some narrative and some system for rewards, usually with experience, levels, or gold; for example, Classcraft [14] lets students personalize characters and real-life powers. These powers are privileges gained by a student's scholarly efforts; these efforts are shown through tasks or quests to achieve within a plot (defeat a final boss or persuade her to make peace with the peasants, or any other idea similar to the fantasy adventures of games like, for example, World of

Warcraft, a massively multiplayer online role-playing game (MMORPG); Dungeons & Dragons, a pen and paper role-playing game (RPG); or Skyrim, a computer role-playing game (CRPG)). To win, they have to complete some academic task. This is the main difference. They are not playing indeed. After they do the task, they receive rewards (experience points), and with enough experience points they level up and obtain powers.

Serious games are those games designed with a formative purpose rather than a playful one. This term appeared in 1970 thanks to Clark C. Abt, an American researcher. He refers to serious games as an approach or simulation that starts from a real situation that develops as a game with an educational intention.

Game-Based Learning refers to complete games that are designed with playful intent and are used in teaching [18,19]. Game-Based Learning creates a fun, motivating, and interactive virtual learning environment, using gaming technologies [20]. It is important to understand differences between gamification and Game-Based Learning. Above it is said that gamification [14] works like an "almost-game". There are several game elements, like story, rewards (experience points, levels, gold, etc.), characters/avatars, and characters' class, but there is not really a game. It is like going to play to Warcraft, Skyrim, Catan, Cluedo, or Dungeons & Dragons (or any other game, these are only examples of very well-known games) and doing everything before playing and after playing but, indeed, not actually playing. In Game-Based Learning [19] you play, because play activity is important, in these cases, for learning.

Within the wide range of Game-Based Learning, we can consider serious games, and also Escape Rooms, as a cooperative activity based in solving puzzles to progress in the plot during a limited amount of time.

The objectives of this paper cover the following:

- Define and understand Escape Rooms, including their origins and influences.
- Identify the advantages, areas of action, and associated problems throughout the implementation of Escape Rooms.
- Identify key issues with Escape Room design and the differences between conventional Escape Rooms and Edu-Escape Rooms.

## 2. Origins

Escape Rooms are one type of Escape Games, which are narrative-based challenges that use puzzles, tasks, and a time limit; these may be table-based paper puzzles or involve trying to open Breakout Boxes. The player and her avatar are the same [21]. These are playful activities that offer an immersive experience. They are usually carried out in groups in a cooperative manner, in which they propose the solving of enigmas or puzzles in order to escape from a fictitious situation [22]. There are variants such as breakouts, in which it is not necessary to escape from a place but rather to manage to open boxes that contain a treasure or the answer to a mystery [23,24]. The origin of these games [25] is not clear, and there are diverse interpretations. Probably one of the most relevant antecedents is the computer game "Origin", from 2006, which achieved a certain success in the USA and Asia. Other computer games from the 80s and 90s, such as graphic adventures, are outstanding milestones. With these precedents, in 2007 the first real Escape Room, outside the virtual environment of a video game, appeared in Japan. Later, these games arrived in the West, specifically in Eastern Europe. However, there are not only precedents in the world of video games. Other activities such as theme parks, films (from detective films based in Agatha Christie or Sherlock Holmes novels to others such as The Name of the Rose, Indiana Jones and the Last Crusade, Cube, Saw, etc.), television shows with live tests, live role-playing games, etc. have had a notable influence on the development of Escape Rooms [26].

Although there is no data in this respect, it is reasonable to consider that teachers who are familiar with Escape Rooms and these activities will have more resources to carry out a correct implementation in the classroom.

When Escape Rooms arrived in Europe, they were configured in a way similar to what we understand today. In large part, this is due to the influence of the Flow Theory [27]. Flow refers to the balance between the challenges faced and the skills with which users must overcome them. Csíkszentmihályi [27] considers that there should be an optimal balance in which learning occurs. When a game maintains a balance between skill and challenge it has flow, and therefore maintains attention and motivation [12].

### 3. Benefits, Experiences, and Design

Although Escape Rooms are a relatively new concept, their educational potential has been studied by several authors such as Moore and Campbell [28] and Clarke et al. [29]. Educational experiences are found in both primary and secondary education, although most of the experiences published in scientific journals are developed in higher education [30–32]. The authors pointed out that there is a significant growth of experiences in recent years, and platforms for sharing experiences such as Breakout EDU [31] are representative of this interest. Regarding the topics covered, the majority correspond to health, welfare, natural sciences, and mathematics, followed by social sciences, information, journalism, and technology. Healthcare and human resources are areas in which Edu-Escape Room experiences have been applied frequently [30,31,33–35]. Along these lines, Schlegel and Radico [36] apply this tool in the interview with resident applicants. They conclude that Edu-Escape Rooms are a useful tool because they allow them to detect the personal skills that require the candidates to develop their profession.

From an institutional perspective, the uses of Escape Rooms are quite varied and include, to cite some examples [31], to recruit students, to help students to get to know institutional services, and to increase students' cataclysm preparedness (like earthquakes). All these activities are intended to be interesting for users who may often be indifferent to the objectives to be achieved, but who find these implementations entertaining and effective.

The creation of these Edu-Escape Rooms requires taking into account a series of aspects, such as type of students, time, material, objectives, etc. [37]. As a counterpart to the questions to be taken into account in their preparation, several researchers [30,31,33,38] point out the advantages of their educational use:

- Improving problem-solving skills
- Encouraging collaborative work
- Learning to think
- Facilitating motivation and learning by doing
- Improving learning immersion
- Developing the imagination
- Enhancing the vision of the whole
- Enjoyment
- Social interaction and communication
- Social relationships and a sense of belonging
- Analytical skills such as critical thinking
- Creativity
- Leadership behaviours
- Can be reused several times with different groups
- As an additional activity for reviewing material

To sum up, social competence, teamwork and collaboration, and motivation seem to be the more frequently named benefits in the literature.

Regarding the challenges and difficulties in this type of experience, the following should be highlighted [30], from more frequently reported to less so:

- Poor evaluation
- Time commitment
- Small sample sizes (for research) *
- Limited resources

- Unbalanced difficulty
- Duration
- Budget limitations
- Large group sizes (more difficult to manage them) *

* We are talking about sample sizes but with different points of view, since the same number of people can be a small sample for research and be too large for a fine management of gamified activity.

Besides, it is very important to take into account the differences between conventional Escape Rooms and educational ones, which must affect design and application (see Table 1).

**Table 1.** Differences between Escape Rooms and Edu-Escape Rooms. Adapted from [31].

| Topic | Conventional Escape Rooms | Edu-Escape Rooms |
|---|---|---|
| Audience | Broad | Specific target group with well-defined learning goals |
| Success rate | Variable | High |
| Puzzles | No need to align with the curriculum | Align with the curriculum |
| Puzzles outcomes | Variable | Numerical or alphabetical codes |
| Spaces | Usually one or several rooms connected | Usually more limited (classroom) |
| Time to set up and clear away | Freer | Limited (academic timetable) |
| Number of users | Usually one team (3–7, average) | A whole class or course (groups of 20–100) |

There can be many difficulties within the variety of Escape Rooms that may happen, but having experience and an appropriate design [29] should reduce the problems that may arise in the execution of the activity. Clarke et al. [29] propose a specific framework for creating educational Escape Rooms, the escapED framework, composed of six areas. These areas are Participants, Objectives, Theme, Puzzles, Equipment, and Evaluation and have been proposed taking into account the review and study of other experiences and works [26,39]. The escapED framework provides teachers and educators an interesting guideline to help develop and apply educational Escape Rooms or Edu-Escape Rooms, as well as other interactive Game-Based Learning activities with a similar design. EscapED framework is indeed a foundation with key elements of entertainment, Game-Based Learning, pedagogic theory, games design, and development processes set up together in one easy-to-follow guideline [29]. This framework provides a holistic approach to developing a learning practice focused on a person-centered approach, highlighting the potential of arts and creativity to design an immersive learning experience in various learning contexts.

Within these considerations on the design of the activity, it takes on special relevance because of the specificity of the puzzles or jigsaws. Nicholson [26,31] identifies different ways of organizing them:

- An open structure (different puzzles can be solved at the same time)
- Sequential structure (solving one puzzle unlocks the next, until the final puzzle can be solved)
- Path-based structure (several paths of puzzles; paths are independent, like the open structure, between them until the end, with a sequential structure inside the path)
- Complex, hybrid structure, which may take the form of a pyramid

Another vital aspect during the Escape Rooms is undoubtedly that of the educators who, in addition to designing the activity, must assume different roles within it: monitoring, guiding, providing hints, and debriefing [26,31].

However, despite the educational potential of the Escape Rooms, we must recognize that there is a lack of studies focusing on exploring their tendencies, affordances, and challenges to education [30–32].

## 4. Conclusions

Some authors [22,30,32,33] underline the growing interest in higher education for Escape Rooms, with a constant increase in publications, focusing especially on universities and health. In Europe, this may be due to the need for methodological changes required by the development of the European Higher Education Area [23]. The fact that there are more higher education experiences published in scientific journals, compared to publications of experiences from other educational levels, may be because higher education professors are more linked to scientific dissemination and research work, so they try to make their innovative educational practices more widely known. An adequate analysis of the impact of the Escape Rooms in other levels would require data that are difficult to obtain, such as courses for non-university teachers, innovation experiences, etc. An example of this is the presence of the Escape Rooms in continuous training activities for teachers, such as El Joc a l'aula (Games in the Classroom), aimed at primary and secondary school teachers in November 2020 with official support from education authorities as part of the DAU (means in Catalan dice) a game festival in Barcelona. Similar training activities have been carried out in other editions of this important festival dedicated to the world of games, and it also includes activities for school groups (https://www.barcelona.cat/daubarcelona/ca/tot-sobre-dau).

Gamification [15–17], Game-Based Learning, serious games, and Escape Rooms provide an attractive perspective for teaching, especially for interventions that encourage student participation and motivation, which are fundamental aspects of learning [5,6]. In this sense, under an educational perspective, we can conclude that the motivation and the student engagement look like the main assets of these proposals. These findings are consistent with previous research that also highlighted their capacity [12,38].

Escape Rooms seem to have an important educational appeal for developing various skills [38] by solving challenges or enigmas collaboratively [22]. One aspect to consider in research could be the profile of the teaching staff involved in these activities, to find out if being familiar with similar games facilitates the approach, even in cases where there is a greater correlation. It could also be interesting to consider which student (or player) profile is more involved in this type of activity (killer, achiever, socializer, or explorer) [12] as each of them may interact differently with the puzzles and find different activities satisfactory. This is a question to consider when designing any playful activity and obviously in the Edu-Escape Rooms.

Although the interest is growing, we have found a limited impact in prestigious journals [30–32], most of the articles being pedagogical experiences. This may be related to the very nature of Escape Rooms, which are more suitable for activities of limited duration that may make it difficult to transfer them to scientific literature. As was mentioned above, it would be interesting to contrast how many related activities of this type are carried out in non-university education.

To be specific, with reference to the objectives set, we found the following:

- Define and understand the Escape Rooms, including their origins and influences.

We defined Escape Rooms as one type of Escape Games, which are narrative-based challenges that use puzzles, tasks, and a time limit. When they are truly educational in intent, their tests are curriculum-related and tend to be used in larger groups with more limited resources than conventional Escape Rooms [26,31].

Their origins and influences cover a very diverse field, including video games, films, mystery novels, role-playing games, improvised theatre, etc. Familiarity with these types of activities can be very useful in the implementation of these activities.

- Identify their advantages, fields of action, and associated problems during their implementation.

An extensive list of the advantages/benefits that have been pointed out in various studies on Escape Rooms was included, including three literature reviews. These include collaborative work, social competence, problem solving, and motivation. On the educational levels, higher education is highlighted, although Escape Rooms are found in primary and secondary education. As we have pointed out previously, this may be related to biases in scientific publications. The fields in which Escape Rooms are most used are clearly those related to health and science, although there is a wide heterogeneity. In relation to the problems, the most common include evaluation, sample size, and resources. It is important to emphasize the differences between conventional and educational Escape Rooms (Table 1), as this poses an additional design challenge.

Undoubtedly, the proposals in which the design is carried out in a planned way, taking into account key elements and paying special attention to how the puzzles are structured, are going to be able to reduce the inconveniences that arise in the application of the Escape Rooms. Without doubt, the experience of the teacher and his/her familiarity with this type of activity is fundamental and will help to develop his/her role in different aspects: monitoring, guiding, providing hints, and debriefing [26,31].

- Identify key issues about its design and the differences between conventional Escape Rooms and Edu-Escape Rooms.

To conclude, we must point out that this kind of activity reflects changes in teaching methodologies and can help to observe and verify the evolution of the educational processes. Edu-Escape Rooms provide education with a challenge-based activity, which, when well designed, can be very attractive to both students and teachers, promoting active and meaningful learning. With sufficient skill and experience, the teaching staff can tackle Escape Rooms that maintain the flow between the challenges and the skills required, integrating this activity into the curriculum and adapting it to organizational issues such as spaces, resources, timetables, and groups. Although the challenge is important and there may be limitations and problems (like limited resources or looking for curriculum, for example), as previously mentioned, the benefits can enrich the educational action, facilitating both social relations and collaborative work or student engagement. Platforms such as Breakout EDU can be of great help for design, as well as different educational tools such as Genially, which provides digital templates. However, probably the most important issue is the design. In this way, the escapED framework with its areas (Participants, Objectives, Theme, Puzzles, Equipment, and Evaluation) is a very useful guideline with a holistic, artistic, emphatical, and creative approach in different learning contexts.

**Author Contributions:** Conceptualization, M.G.-d.-P. and S.G.-M.; writing—original draft preparation, M.G.-d.-P. and S.G.-M.; writing—review and editing, R.B. and V.A.-G. All authors have read and agreed to the published version of the manuscript.

**Funding:** This research received no external funding.

**Acknowledgments:** Tony Fuentes, translation reviewer.

**Conflicts of Interest:** The authors declare no conflict of interest.

**Entry Link on the Encyclopedia Platform:** https://encyclopedia.pub/3481.

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
