# Peer review of "Edu-Escape Rooms"

_encyclopedia, doi:10.3390/encyclopedia1010004_

Round 1

Reviewer 1 Report

The submitted manuscript as Entry deals with edu-escape rooms, which is related to the main focus of this journal. However, it has still some flaws I can find out, especially manuscript formatting, main text and language issue.

First, manuscript is not prepared according to the authors’ guidelines of the journal. Especially, the references’ formatting is not followed by the journal guidelines. Also, there are many incorrect uses of acronym. Please revise the whole text along with mentioned comments.

Second, in the section of benefits, it would be encouraged to write a more academic writing that the current form is not recommendable for the journal of Encyclopedia. Also, vverall paper does a poor job in presenting the details of the research and contextualing the study within the international literature related to the topic. There are many publications only in a specified region or country that will not give an international aspect for the manuscript.

Finally, particularly, language is rough and presentation is poor. Please carefully revise the main text with a help of native speakers.

Author Response

Answers to reviewers

20th/November/2020

Manuscript Encyclopedia-991290

Title: Edu-Escape rooms

We thank all the reviewers for their thorough comments and final recommendations that helped to improve the manuscript. Thanks to them, we think that the paper is strongly improved, especially regarding the effectiveness in communication. We hope that the current version answers their concerns.

In the following, the detailed comments from reviewers and our responses are reported. The suggestions have been incorporated into the manuscript. Likewise, we have highlighted these changes in the article. Moreover, references have been incorporated in order to answer reviewers’ comments.

Reviewer 1

R1.1

The manuscript is not prepared according to the authors’ guidelines of the journal. 

Answer

We have followed the template and authors’ guidelines that are on the website; https://www.mdpi.com/journal/encyclopedia/instructions. Please, if we are misinterpreting some recommendations, let us know.

R1.2

The references’ formatting is not followed by the journal guidelines. Also, there are many incorrect uses of acronyms.

Answer

We have reviewed the format of the references according to the American Chemical Society style as well as the reference list and citation style guide provided on MDPI website; https://www.mdpi.com/authors/references (and ACS style). Once again, your feedback and corrections are welcome if the indications are not interpreted correctly by us.

As the extent of the text is not too long and there are not excessive concept repetitions, we have deleted the acronyms to facilitate the reading.

R1.3

Write a more academic writing.

Answer

It has been rewritten some parts of the paper following the indications for Ecyclopedia’s entries (https://encyclopedia.pub/guideline) in order to have a more academic style. 

R1.4

The paper does a poor job in presenting the details of the research and contextualizing the study within the international literature related to the topic

Encyclopedia's guideline remarks that the length of the entries is around 1000 words and the same document show that one of the possibilities for the Topic review is "(2) present the conclusions of one or several research projects on a single, specific topic, or a general review of a research field. Results can relate to a single field, a research group, or a collective of researchers". This paper has around 1200 words in which it has been summarized and reviewed the findings of the topic from more than 35 papers. Of the studies reviewed more than 83% are in English and most of them are not Hispanic-American countries or authors, so we do not consider that the entry has not a solid international review perspective.

R1.4

There are many publications only in a specified region or country that will not give an international aspect for the manuscript.

Answer

It has been increased the number of international references to have more representation on an international scale. Of the studies reviewed more than 60% come from non-Hispanic-American countries and authors, so we consider that the entry has a wide and solid international review perspective. 

R1.5

The language is rough and presentation is poor. Please carefully revise the main text with a help of native speakers.

Answer

As the entry was written by non-native-English authors, this review we have counted with the collaboration of a native English speaker and writer that revise and correct it. We hope the result sounds better, more academic and more appropriate. 

Reviewer 2 Report

The entry, globally, seems to me to be correct and meets the usual requirements in  Encyclopedia. However, we believe that the references include a large number of sources from the Latin American area (13 of 22), which may not be representative on an international scale. A review of this would be desirable.

Author Response

Answers to reviewers

20th/November/2020

Manuscript Encyclopedia-991290

Title: Edu-Escape rooms

We thank all the reviewers for their thorough comments and final recommendations that helped to improve the manuscript. Thanks to them, we think that the paper is strongly improved, especially regarding the effectiveness in communication. We hope that the current version answers their concerns.

In the following, the detailed comments from reviewers and our responses are reported. The suggestions have been incorporated into the manuscript. Likewise, we have highlighted these changes in the article. Moreover, references have been incorporated in order to answer reviewers’ comments.

Reviewer 2

R2.1

Large number of sources from the Latin American area (13 of 22).

Answer

It has been increased the number of international references related to the topic (+14) to have more wide and solid on an international perspective. Now, more than 60% of the studies reviewed come from non-Hispanic-American countries and authors.

Reviewer 3 Report

I recommend the authors to better define their objective and to explain more coherently which are concrete, obvious advantages of using this method in education: for what level of education is the method applicable, what are the necessary competencies for students and trainers, with what results implemented so far etc.
Topic is very interesting and I think it deserves developed.

Author Response

Answers to reviewer 3

12/December /2020

Manuscript Encyclopedia-991290

Title: Edu-Escape Rooms

Estimated reviewer,

We thank all the reviewers, from both first and second round, for their thorough comments and final recommendations. They have definitely helped to improve the manuscript. We hope that the current version answers their concerns.

In the following, the detailed comments are addressed. To properly address these suggestions, the general comment has been divided into 5 comments are presented in black and our responses are reported in blue. The suggestions have been incorporated into the manuscript.

Kind regards,

Authors

Reviewer 3

"I recommend the authors to better define their objective and to explain more coherently which are concrete, obvious advantages of using this method in education: for what level of education is the method applicable, what are the necessary competencies for students and trainers, with what results implemented so far etc. Topic is very interesting, and I think it deserves developed."

R3.1

"I recommend the authors to better define their objective…

Answer

Clear objectives have been incorporated to the manuscript as bullet points in order to clarify them (page 2, lines 68-72). Moreover, we have given answer to them in the conclusions (pages 5 and 6, lines 225-266).

R3.2

…and to explain more coherently which are concrete, obvious advantages of using this method in education…

Answer

We have listed and extended the advantages, the challenges and the difficulties which this method has in education. We have as well highlighted the advantages most cited in literature and summarized them in consonance with our experience (pages 3 and 4, lines 120-152).

R3.3

…for what level of education is the method applicable…

Answer

The reviewed literature shows that this is a method with a broad application in education. There are experiences in Primary Education, Secondary Education and High School, although the majority of experiences that have been published in scientific journals have been developed in Higher Education (page 2, lines 101-107).

R3.4

…what are the necessary competencies for students and trainers…

Answer

We assume (and included in the paper) that the familiarity and experience with Edu-Escape Rooms or similar activities, play a significant role in their success (page 2, lines 90-92; page 5, lines 212-218). Other significant competencies required are related to their design (instructional) and the method training that determines trainer skills to monitor, guide and give feedback (p.4, lines 181-183).

R3.5

… with what results implemented so far etc.”

Answer

Several sections have been improved in relation to this comment. In this sense, some of the most outstanding results are related to the replication or simulation of institutional experiences, the promotion of the image of the institution, etc.

However, we recognize that there is not concluding evidence about the results of the application of Edu-Escape Rooms. Although, preliminary studies show a high educational potential in relation to the description of personality traits, skills development, etc.

(page 3, lines 112-116; page 5, lines 186-188 and lines 224-236; page 6, lines 237-266).

Reviewer 4 Report

The authors provide a well-sustained entry concerning a  playful alternative to the traditional learning methodologies, the Edu-escape rooms. In fact, the proliferation of games has become eminent in our society, and one should take advantage of it. Hence, this entry establishes a good starting point for those who are interested in the subject, since it is supported by important factual articles. The document is easy to read/follow, which means that little spelling check is required.

Author Response

Estimated reviewer,

We appreciate the interest that reviewers have taken in our manuscript.

We thank all the reviewers, from both first and second round, for their thorough comments and final recommendations.

They have definitely helped to improve the manuscript.

After last round you did not ask for any changes, and no further comments are included here. 

Kind regards

Round 2

Reviewer 1 Report

First of all, I appreciate to the authors for making efforts to carry out the changes by the referees. I think the authors did a good job in clarifying the queries that this manuscript is substantially improved.

Author Response

(The authors gave the same response as above.)

Reviewer 2 Report

The recommended changes have been made.

Author Response

(The authors gave the same response as above.)

Reviewer 3 Report

The authors have made efforts to improve the work and brought new information relevant to the topic discussed.This demonstrates concern for the theme addressed but also to comply with conditions of a scientific article.